# Integrated Analysis of Phagocytic and Immunomodulatory Markers in Cervical Cancer Reveals Constellations of Potential Prognostic Relevance

**DOI:** 10.3390/ijms25169117

**Published:** 2024-08-22

**Authors:** Angel Yordanov, Polina Damyanova, Mariela Vasileva-Slaveva, Ihsan Hasan, Stoyan Kostov, Velizar Shivarov

**Affiliations:** 1Department of Gynecologic Oncology, Medical University-Pleven, 5800 Pleven, Bulgaria; 2Department of General and Clinical Pathology, Heart and Brain Center of Clinical Excellence, 5800 Pleven, Bulgaria; poldamdim@abv.bg; 3Research Institute, Medical University Pleven, 5800 Pleven, Bulgaria; sscvasileva@gmail.com (M.V.-S.); drstoqn.kostov@gmail.com (S.K.); vshivarov@abv.bg (V.S.); 4Department of Breast Surgery, Shterev Hospital, 1000 Sofia, Bulgaria; 5Department of Obstetrics and Gynecology, University Hospital “Sofiamed”, 1750 Sofia, Bulgaria; ihsan_hasanov@abv.bg; 6Department of Gynecology, Hospital “Saint Anna”, Medical University—“Prof. Dr. Paraskev Stoyanov”, 9002 Varna, Bulgaria

**Keywords:** cervical cancer, immunity, CD68, CD47, prognosis

## Abstract

Despite improvements in vaccination, screening, and treatment, cervical cancer (CC) remains a major healthcare problem on a global scale. The tumor microenvironment (TME) plays an important and controversial role in cancer development, and the mechanism of the tumor’s escape from immunological surveillance is still not clearly defined. We aim to investigate the expression of CD68 and CD47 in patients with different histological variants of CC, tumor characteristics, and burden. This is a retrospective cohort study performed on paraffin-embedded tumor tissues from 191 patients diagnosed with CC between 2014 and 2021 at the Medical University Pleven, Bulgaria. Slides for immunohistochemical (IHC) evaluation were obtained, and the expression of CD68 was scored in intratumoral (IT) and stromal (ST) macrophages (CD68+cells) using a three-point scoring scale. The CD47 expression was reported as an H-score. All statistical analyses were performed using R v. 4.3.1 for Windows. Infiltration by CD68-IT cells in the tumor depended on histological type and the expression of CD47. Higher levels of the CD47 H-score were significantly more frequent among patients in the early stage. Higher levels of infiltration by CD68-ST cells were associated with worse prognosis, and the infiltration of CD68-IT cells was associated with reduced risk of death from neoplastic disease. TME is a complex ecosystem that has a major role in the growth and development of tumors. Macrophages are a major component of innate immunity and, when associated with a tumor process, are defined as TAM. Tumor cells try to escape immunological surveillance in three ways, and one of them is reducing immunogenicity by the overexpression of negative coreceptors by T-lymphocytes and their ligands on the surface of tumor cells. One such mechanism is the expression of CD47 in tumor cells, which sends a “don’t eat me” signal to the macrophages and, thus, prevents phagocytosis. To our knowledge, this is the first study that has tried to establish the relationship between the CD47 and CD68 expression levels and some clinicopathologic features in CC. We found that the only clinicopathological feature implicating the level of CD68 infiltration was the histological variant of the tumor, and only for CD68-IT–high levels were these observed in SCC. High levels of CD47 expression were seen more frequently in pT1B than pT2A and pT2B in the FIGO I stage than in the FIGO II and III stages. Infiltration by large numbers of CD68-IT cells was much more common among patients with a high expression of CD47 in tumor cells. A high level of infiltration by CD68-ST cells was associated with a worse prognosis, and a high level of infiltration by CD68-ST cells was associated with a lower risk of death from cancer.

## 1. Introduction

The incidence of cervical cancer (CC) is decreasing in developed countries because of vaccination and screening programs. Unfortunately, in the rest of the world, it remains a major healthcare problem. Worldwide, CC ranks fourth, both in terms of morbidity and mortality from oncological diseases among women [1], with over 604,000 new cases and over 340,000 deaths registered in 2020 [2]. This translates to a 5.7% increase in new cases and an 8.9% increase in mortality in 2020 compared to 2018 [1].

The tumor microenvironment (TME) is a complex, dynamically evolving heterogeneous ecosystem consisting of cellular and non-cellular components. It has a major role in the development of oncological processes. Depending on the function they perform in antitumor immune response, these cells can be divided into two groups—immunosuppressive and immunostimulating [3]. The interaction of these cells with each other and with tumor cells leads to the activation or inhibition of various immune mechanisms that can suppress or promote the development and progression of CC. The immune response is determined by the interaction of the adaptive and innate immune systems, and their role is not well studied in CC.

CD68 is a transmembrane glycoprotein that is expressed by the monocyte cell line (mainly by macrophages) and by osteoclasts. Tumor-associated macrophages (TAMs, CD68+) are a major part of innate immunity and are elevated in many cancers. They influence the activity of other immune cells, thereby creating a pro- or antitumor microenvironment [4].

Cluster of Differentiation 47 (CD47), also called the integrin-associated protein (IAP), is a transmembrane immunoglobulin encoded by the CD47 gene [5] and found on the surface of many different types of cells in the human body. It protects cells from destruction by circulating macrophages. The CD47 protein combines in a strong signaling complex with another signaling and regulatory protein, SIRPα, which also produces the “don’t eat me” signal [6]. CD47 is highly expressed in various tumor types and is associated with poor prognosis.

The aim of this study was to investigate the expression of CD68 and CD47 in patients with different histological variants of CC at different stages and to analyze the relationship between their expression and the FIGO (International Federation of Gynecology and Obstetrics) stage, T and N status, and surveillance.

## 2. Results

We decided to evaluate the infiltration of TAMs by measuring the number of CD68+ cells infiltrating the tumor tissue and its potential correlation with any demographic and clinical characteristics of the patients. We defined two measures for TAM, defined as intratumoral CD68+ cells (CD68-IT) and stromal CD68+ cells (CD68-ST). As shown in Table 1, the only significant correlation difference in the level of infiltration by TAMs was the unequal distribution of CD68-IT per histological subtype. Patients who were assessed as having a high infiltration by CD68-IT cells in the tumor were identified only in the group of squamous cell carcinoma (SCC) patients but not among the adenocarcinoma (AC) and adenosquamous carcinoma (ASC) patients (Fisher’s test *p* < 0.0001), (Table 1, Figure 1A).

We additionally evaluated whether the check-point molecule CD47, which is considered a negative regulator of phagocytosis, showed any differences in its expression in tumor cells between patients’ subgroups. Of note, tumors expressing higher levels of the CD47 H-score were significantly more frequent among patients with T1 (pT1b) tumors than in those with locally advanced tumors (pT2a and pT2b) Fisher’s test *p* < 0.0001) (Table 1, Figure 1B). There is a significant difference in CD47 expression by FIGO stage (Fisher’s test *p* = 0.0012) (Table 1, Figure 1C).

Another obvious question is whether infiltration by TAMs is associated with any differences in the expression of CD47 in tumor cells. Of note, we observed one such significant correlation (Table 1). Infiltration by high numbers of CD68-IT cells was much more frequent among patients with high expressions of CD47 in tumor cells (*p* = 0.006) (Figure 1D)

Led by the initial observations of some clinical and pathological correlations of infiltration by TAMs and the expression of CD47, we decided to investigate the correlation of these markers in a multivariate fashion using linear models. As shown in Figure 2A, infiltration by CD68-IT cells remained independently associated with SCC histology (*p* = 0.001), but the association with the CD47 H-score lost statistical significance (*p* = 0.058). There was only a tendency for higher CD68-IT infiltration in patients with pT2a and pT2b tumors compared to patients with pT1b (Figure 2A).

Performing analogous multivariate linear model analysis for the infiltration of CD68-ST cells did not reveal any significant associations (Figure 2B). Finally, in multivariate linear models, the CD47 H-score appeared to be significantly lower in patients with both pT2a and pT2b tumors than in those with pT1b. 

The CD47 H-score showed a strong tendency to associate with higher levels of infiltrating CD68-ST cells, but this had marginal statistical significance (*p* = 0.05) (Figure 2C).

Infiltration by immune cells and the expression of co-stimulatory or inhibitory molecules by tumor cells clearly play pivotal roles in anti-cancer immune response and its evasion. Our observations that there was a tendency for the correlation of infiltration by CD68 cells and the expression of inhibitory molecules such as CD47, as well as the fact that such correlations might be influenced by histological subtype and tumor stage, led us to question whether these markers had any prognostic implications.

Univariate survival analysis using the log-rank test showed that the level of infiltration by CD68-IT did not correlate with survival (*p* = 0.72) (Figure 3A). However, high levels of infiltration by CD68-ST cells were associated with a worse prognosis (*p* = 0.049) (Figure 3B). Finally, the levels of expression of CD47 did not define subgroups of patients with different prognoses (*p* = 0.16) (Figure 3C).

We then fitted a multivariate Cox model for overall survival with these IHC biomarkers as covariates alongside age, T- and N-stage, and histological subtype. As shown in Figure 3D, only age at diagnosis remained an independent prognostic factor for shorter overall survival (*p* = 0.005).

This led us to investigate whether this association was affected by death from other causes and not by underlying neoplastic disease. To this end, we also performed a competing risk regression analysis, assuming death from neoplastic disease and death due to other causes as competing risks. As shown in Figure 4A, there was no difference between the cumulative risk functions for the risk of dying from either neoplastic disease or any other cause. We finally fitted the subdistribution hazard model, known also as the Fine-Gray model, to analyze the cumulative incidence of dying from a neoplastic disease. As shown in Figure 4B, it appears that infiltration by CD68-IT cells has independent statistical significance of being associated with reduced risk of death from neoplastic disease.

## 3. Discussion

TME is a complex ecosystem that has a major role in the growth and development of tumors [7,8]. It consists of various cellular and extracellular components. Cellular components include tumor-infiltrating immune cells (lymphoid and myeloid cells) and stromal cells—tumor-associated fibroblasts and endothelial cells and tumor cells [9,10,11,12]—and extracellular components include various cytokines, hormones, the extracellular matrix and growth factors [13].

Macrophages are a major component of innate immunity and, when associated with a tumor process, are defined as TAM. They are divided into two phenotypes—M1 and M2 macrophages [14]. M1 inhibits tumor growth through the synthesis of proinflammatory cytokines, such as tumor necrosis factor-α (TNF-α), IL-1, IL6, IL-12, IL-23, and reactive nitrogen and intermediate oxygen compounds, thus ensuring the phagocytosis of defective cells [15]. M2 stimulates tumor growth and invasion through the secretion of IL-4, IL-13, IL-10, vitamin D3, and glucocorticoids [16,17]. CD68 is a marker for both phenotypes.

Tumor cells try to escape immunological surveillance in three ways: the loss of antigenicity; the loss of immunogenicity; and the modulation of an immunosuppressive microenvironment [18]. Immunogenicity is reduced by the overexpression of negative coreceptors of T-lymphocytes and their ligands on the surface of tumor cells. One such mechanism is the expression of CD47 in tumor cells, which sends a “don’t eat me” signal to macrophages and thus prevents phagocytosis [19,20].

CD47 is known to have a very important role in oncogenesis in many malignant diseases by stimulating the growth, invasion, and migration of cancer cells [21]. In some cancers, CD47 expression is associated with advanced stage at diagnosis, lymphogenous metastases, and relapse [19,21].

Although the role of TAMs in cancer progression is well studied, the knowledge of its particular place in CC is limited [22]. It is known that TAMs play a determinative role in CC development and progression [22]. The evolution of CC starts with an inflammatory process that activates different signals, such as the nuclear factor kappa-light-chain-enhancer of activated B cells (NF-κB) and hypoxia-inducible factor (HIF-1α) with a signal transducer and activator of transcription 3 (Stat3) in tumor cells which cause the production of IL-10 or transforming growth factor beta (TGF-β) in the TME by TAMs [23,24]. TAMs play a crucial role in CC angiogenesis [25]. They are cumulated in hypoxic TMEs [26] by different chemoattractants such as CCL2, CCL5, or colony-stimulating factor 1 (CSF1) [27]. An increase in the levels of TAMs stimulates angiogenesis by producing proangiogenic factors like the vascular endothelial growth factor (VEGF), fibroblast growth factor (FGF), platelet-derived growth factor (PDGF), MMPs, IL-1, IL-8, TNF-α, and nitric oxide (NO) [28,29]. This leads to the raised potential of tumoral inflammation and lymphangiogenesis in CC [30,31]. Chen et al. suggested that an increase in the number of TAMs determines metastasis and poor prognosis in CC [32]. TAMs have a main role in the invasion and metastasis of CC cells [33], where the higher levels determine regional and distant metastasis and worsened prognosis [17,34,35]. TAMs have an immunosuppressive effect in TME in CC by inhibiting T cell proliferation and promoting FOXP3 (forkhead box P3)+ Treg subsets [36,37]. TAMs have a serious impact on cancer stem cells (CSCs)–they stimulate the evaluation of more chemo-resistant and invasive CSCs in some cancers [38,39]. Nevertheless, the relation of CSC–TAM has not been adequately examined in CC [22]. From the literature, it is known that higher concentrations of TAMs are associated with poor prognosis [40] and tumor progression [41].

We found only one study, performed by the same team [42], which presented the role of CD47 studied by IHCSo to our knowledge, this is the first study that has tried to establish a relationship between the CD47 and CD68 expression levels and some clinicopathologic features in CC. In the literature, there are similar isolated reports for other neoplastic diseases. Chen et al. reported that a high expression of both markers is an independent prognostic factor associated with poor prognosis in all types of breast cancer [43]. Similar results were reported by Yuan et al. [44]. According to Fu et al., CD47 and CD68 expression levels are an excellent predictive survival factor of non-small-cell lung cancer patients [45]. However, Semiz et al. find no correlation between the levels of expression for these two markers and survival rate in early-stage prostate cancer [46]. We should note that none of these authors divide CD68 into CD68-IT and CD68-ST, and no one has considered their specific role in cancer.

We found no correlation between the infiltration levels of CD68-IT and CD68-ST and the clinicopathological characteristics of the patients for age, FIGO stage, T, and N status. An exception is the histological type of the tumor, where patients with high infiltration of CD68-IT cells were identified only in the group of patients with SCC but not among patients with AC and ASC (*p* < 0.0001)

The results are similar when examining the expression of the CD47 H-score, where there is no relationship between the degree of expression and age, histological type, and N status. What is striking is that tumors with higher CD47 H-score levels were significantly more common among patients with T1 (pT1b) tumors than among locally advanced (pT2a and pT2b) (*p* < 0.0001). This observation was also confirmed by the fact that CD47 expression is higher in FIGO stage I than in stages II and III (*p* = 0.0012). And here, it could be speculated that at the beginning of tumorigenesis, the tumor escapes immunological control through the expression of CD47; that is, it becomes invisible to macrophages, and, subsequently, other mechanisms are involved.

These results led us to wonder whether there is a relationship between infiltration by TAMs and CD47 expression in tumor cells. And indeed, there is such a relationship as infiltration by numerous CD68-IT cells was much more frequent among patients with a high expression of CD47 in tumor cells (*p* = 0.006). In multivariate analysis, this relationship did not persist (*p* = 0.058). There was only a trend for higher CD68-IT infiltration in patients with pT2a and pT2b tumors compared to patients with pT1b. There was also a strong trend for the CD47 H-score to be associated with higher levels of CD68-ST cell infiltration, but this showed marginal statistical significance (*p* = 0.05).

We found that levels of CD68-IT infiltration and CD47 expression did not correlate with survival (*p* = 0.72 and *p* = 0.16, respectively), while a high level of CD68-ST cell infiltration was associated with worse prognosis (*p* = 0.049). However, we found that high levels of infiltration by CD68-IT resulted in a lower risk of death from neoplastic disease (*p* = 0.043).

## 4. Materials and Methods

This is a retrospective study performed on paraffin-embedded tumor tissues from 191 patients with CC diagnosed between 2014 and 2021 in the university hospitals of Medical University-Pleven, Bulgaria. We used samples available in the hospital’s pathology archive that contained a sufficient amount of tumor tissue and where the study would not endanger their depletion or damage. Permission for the study was obtained from the Ethics Committee (number 656/29 June 2021). Clinico-morphological data for the patients were obtained from the electronic database of the Department of Gynecologic Oncology, where all patients underwent primary surgery.

### 4.1. Patients Characteristics

We collected data on patient age and tumor stage at diagnosis according to TNM (tumor-node-metastasis) classifications (by the American Joint Committee on Cancer staging manual, 8th edition) and FIGO classification 2009 (Table 2).

In the FIGO III stage, only patients with lymph node metastases were included (FIGO IIIC) according to the current guidelines for the treatment of CC primary surgery, which is not recommended for patients in stages FIGO IIIA and FIGO IIIB. Some of these results are reported in two previous articles [42,47].

### 4.2. Immunohistochemical Examination

One histological slide with hematoxylin-eosin staining was selected for all patients. Slides for immunohistochemical (IHC) evaluation were obtained from corresponding formalin-fixed, paraffin-embedded (FFPE) tissue specimens. From FFPE tissue samples, 2–4 µm thick tissue sections were prepared and placed on adhesive slides. Each of the 191 studied patients was tested with each of the two primary antibodies (for CD47, we used clone SP279, Rb, dilution 1:100, Abcam, UK, pre-diluted and ready-to-use alongside CD68-clone PG-M1, Mo, Dako/Agilent, Glostrup, Denmark). The EnVision™ FLEX, High pH (Link), DAKO Antibody Detection System, and an Auto-stainerLink 48 automated system, DAKO, were used.

All working procedures in conducting the IHC analysis were carried out according to the protocols for the respective antibodies of the manufacturer company. In each staining, external control tissues were used to establish the functionality of the staining reagents, to assess the quality of the staining reaction, to determine the expression pattern of the antibodies used, and to optimize the IHC work procedures prior to their application to the studied cases.

CD68 expression was scored in intratumoral (IT) and stromal (ST) macrophages using a three-point scoring scale. IT macrophages are those with direct contact with tumor cells, mainly in the tumor nests or in their periphery, and ST ones–those in the tumor stroma without direct contact with neoplastic cells. The scoring methodology was described in detail in our previous report (Figure 5A–D) [47]. The mean value of the reported results was taken as a cut-off (Table 3).

Currently, there is no accepted scoring system for evaluating CD47 expression; therefore, we decided to report it as an H-score since, in our previous publication, we found that this method was the most appropriate (Figure 5E–H) [42]. The expression level was separated by the median value of the H-score: low (with H-score ≤ 74) or high (with H-score > 74). If there were <1% positive cells with H-score = 0, it was considered to be a negative result.

### 4.3. Statistical Analyses

All IHC markers were cross-tabulated versus the main clinical and pathological parameters such as age (defined as a categorical variable with a cut-off of 50 years), T stage, N stage, and histological subtype. The obtained contingency tables were analyzed using Fisher’s exact test because of the low numbers of some cells in the contingency tables. *p*-values below 0.05 were considered significant. Multivariate linear regressions for all three IHC phagocytosis markers were performed using all available clinical and pathological parameters as co-variables. The survival curves for subgroups of patients defined by IHC markers were built using the Kaplan–Meier method and analyzed for statistical significance using the log-rank test. Kaplan–Meier curves were plotted and analyzed using the survminer package for R 0.4.9. The multivariate Cox model was fitted to evaluate overall survival utilizing all clinical and pathological parameters as co-variables. It was performed using the survival package for R. Cumulative incidence functions for risk of dying from either neoplastic disease or any other cause were estimated using the Kaplan–Meier method and plotted using the survminer package for R. The multivariate Fine-Gray model for estimating the effect of covariates on the risk of dying from neo-plastic disease was performed using the risk regression package for R. Results from multivariate statistical models are presented using either forest model or forest plot packages for R. All statistical analyses were performed using R v. 4.3.1 for Windows.

## 5. Conclusions

We found that the only clinicopathological feature implicating the level of CD68 infiltration was the histological variant of the tumor and only for CD68-IT; as such, high levels were only observed in SCC. High levels of CD47 expression were seen more frequently in pT1B than pT2A and pT2B and in the FIGO I stage compared to stages FIGO II and III, respectively. Infiltration by large numbers of CD68-IT cells was much more common among patients with a high expression of CD47 in tumor cells. A high level of infiltration by CD68-ST cells was associated with worse prognosis, and a high level of infiltration by CD68-ST cells was associated with a lower risk of death from cancer.

## Figures and Tables

**Figure 1 ijms-25-09117-f001:**
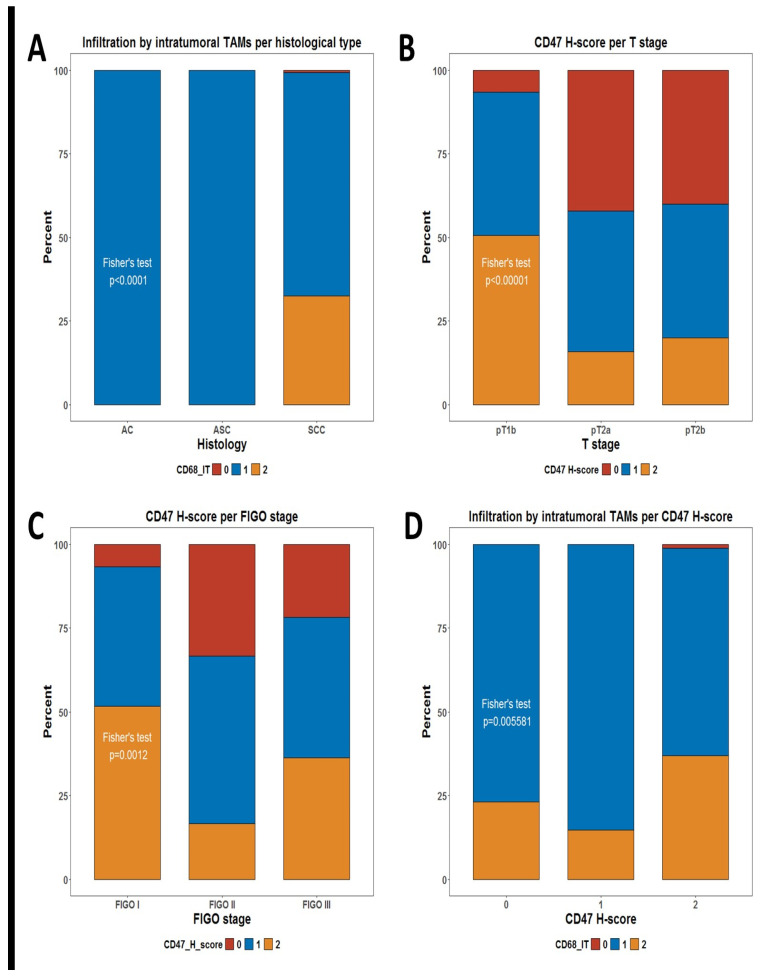
Stacked bar plots summarizing distributions of some phagocytic markers. (**A**) Distribution of levels of infiltration for intratumoral TAMs per histological subtype; (**B**) distribution of CD47 H-score groups per T stage; (**C**) distribution of CD47 H-score groups per FIGO stage; and (**D**) distribution of levels of infiltration for intratumoral TAMs per CD47 H-score groups. *p*-values are from Fisher’s test, as shown in Table 1.

**Figure 2 ijms-25-09117-f002:**
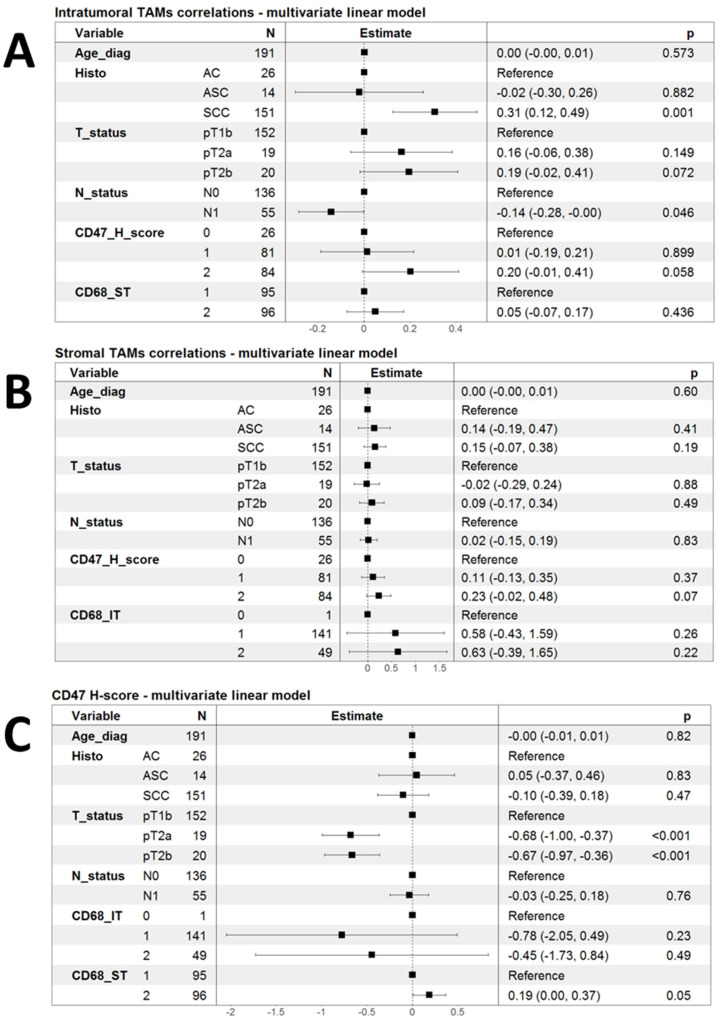
Forest plots summarizing multivariate linear models for the correlation of phagocytic markers. (**A**) Linear model of infiltration by intratumoral TAMs as a function of clinical variables and the other two markers of phagocytosis; (**B**) a linear model of infiltration by stromal TAMs as a function of clinical variables and the other two markers of phagocytosis; and (**C**) linear model of infiltration by CD47 H-score as a function of clinical variables and the other two markers of phagocytosis.

**Figure 3 ijms-25-09117-f003:**
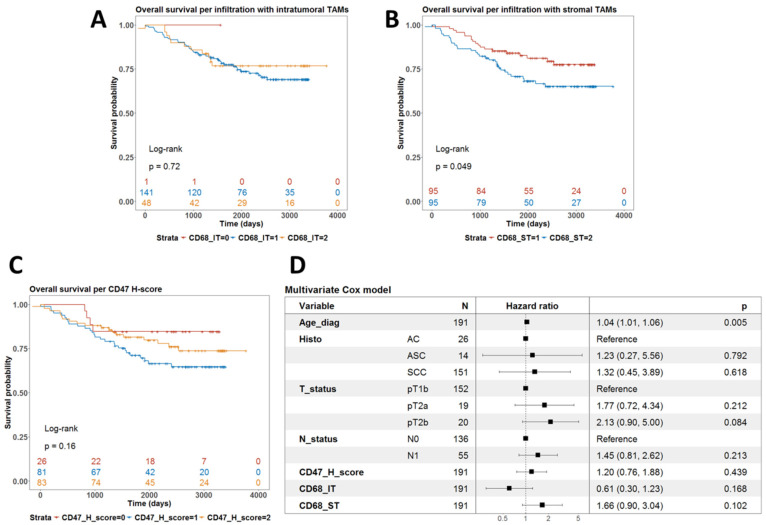
Analysis of the prognostic value of phagocytic markers in terms of overall survival. (**A**) Univariate survival with the level of infiltration by intratumoral TAMs. *p*-value is from the log-rank test. (**B**) Univariate survival with the level of infiltration by stromal TAMs. *p*-value is from the log-rank test. (**C**) Univariate survival with CD47 H-score. *p*-value is from the log-rank test. (**D**) Forest plot summarizing the results from a multivariate Cox model for overall survival using clinical covariates and all three phagocytic markers, which were treated as continuous variables.

**Figure 4 ijms-25-09117-f004:**
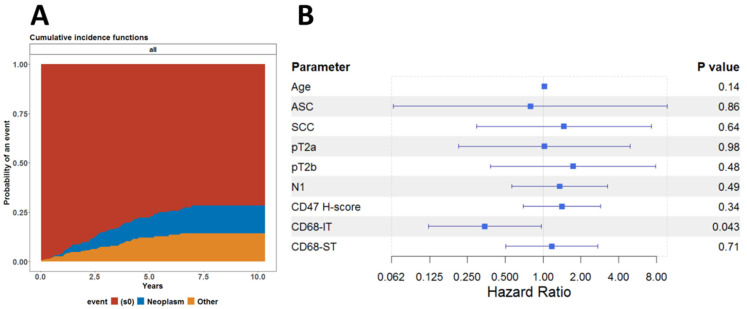
Competing risk regression analysis. (**A**) Cumulative incidence functions for risk of dying from either neoplastic disease or any other cause for all patients. (**B**) Forest model summarizing the results of the Fine-Gray model fitted to evaluate the effect of all clinical and pathological variables on the cumulative incidence of death due to neoplastic disease.

**Figure 5 ijms-25-09117-f005:**
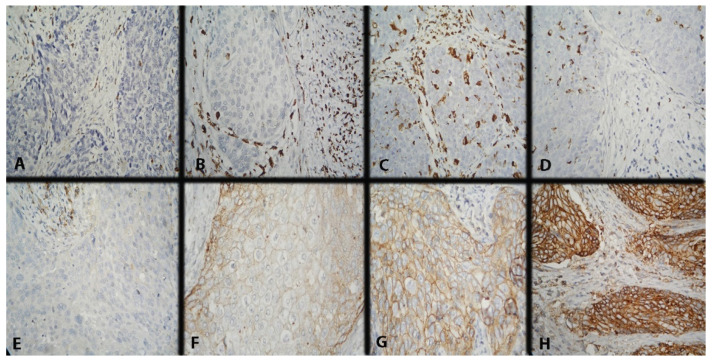
Microscopic evaluation of IHC-stained CD68+ macrophages (colored brownish) and IHC expression model of CD47 in CC. Magnification ×400. (**A**) Low intratumoral and stromal CD68 expression; (**B**) low intratumoral and high stromal CD68 expression; (**C**) high intratumoral and stromal CD68 expression; (**D**) high intratumoral and low stromal CD68 expression; (**E**) missing membrane positivity of CD47 expression; (**F**) weak membrane positivity of CD47 expression; (**G**) average membrane positivity of CD47 expression; and (**H**) strong membrane positivity of CD47 expression.

**Table 1 ijms-25-09117-t001:** Distribution of CD68- and CD47-positive cells per subgroups of patients.

	CD68IT					CD68ST				CD47				
	0	1	2		*p*-value	1	2		*p*-value	0	1	2		*p*-value
T stage				Total				Total					Total	
T1a	1	114	37	152	0.6806	75	77	152	0.7251	10	65	77	152	1.70 × 10^−6^
T2a	0	14	5	19		11	8	19		8	8	3	19	
T2b	0	13	7	20		9	11	20		8	8	4	20	
Total	1	141	49	191		95	96	191		26	81	84	191	
N stage														
N0	0	99	37	136	0.2716	68	68	136	1	14	58	64	136	0.09195
N1	1	42	12	55		27	28	55		12	23	20	55	
Total	1	141	49	191		95	96	191		26	81	84	191	
FIGO stage														
FIGO I	0	87	31	118	0.4435	58	60	118	0.9047	8	49	61	118	0.0012
FIGO II	0	12	6	18		10	8	18		6	9	3	18	
FIGO III	1	42	12	55		27	28	55		12	23	20	55	
Total	1	141	49	191		95	96	191		26	81	84	191	
Histology														
AC	0	26	0	26	3.41 × 10^−5^	17	9	26	0.2297	2	14	10	26	0.514
ASC	0	14	0	14		7	7	14		1	8	5	14	
SCC	1	101	49	151		71	80	151		23	59	69	151	
Total	1	141	49	191		95	96	191		26	81	84	191	
Age														
≤50 years	1	71	22	94	0.5631	49	45	94	0.5638	11	39	44	94	0.6619
>50 years	0	70	27	97		46	51	97		15	42	40	97	
Total	1	141	49	191		95	96	191		26	81	84	191	
CD68IT														
0	1	0	0	1	NA	1	0	1	0.1851	0	0	1	1	0.005581
1	0	141	0	141		74	67	141		20	69	52	141	
2	0	0	49	49		20	29	49		6	12	31	49	
Total	1	141	49	191		95	96	191		26	81	84	191	
CD68ST														
1	1	74	20	95	0.1851	95	0	95	NA	16	44	35	95	0.1246
2	0	67	29	96		0	96	96		10	37	49	96	
Total	1	141	49	191		95	96	191		26	81	84	191	
SD47 H score														
0	0	20	6	26	0.005581	16	10	26	0.1246	26	0	0	26	NA
1	0	69	12	81		44	37	81		0	81	0	81	
2	1	52	31	84		35	49	84		0	0	84	84	
Total	1	141	49	191		95	96	191		26	81	84	191	

AC—adenocarcinoma; ASC—adenosquamous carcinoma; and SCC—squamous cell carcinoma.

**Table 2 ijms-25-09117-t002:** Characteristics of patients.

Patients’ Characteristics	N	(%)
FIGO I	118	61.8
FIGO II	18	9.4
FIGO III	55	28.8
T1b *	152	79.6
T2a	19	9.9
T2b	20	10.5
N0	136	71.2
N1	55	28.8
squamous-cell carcinoma	151	79.1
adenocarcinoma	26	13.6
adenosquamous carcinoma	14	7.3
Total	191	100

* Includes patients at stage T1b1, T1b2, and T1b3.

**Table 3 ijms-25-09117-t003:** Reporting of CD68 expression in intratumoral and stromal macrophages of CD47 expression in tumor cells.

Levels of Positive Cells	Intratumoral TAMs, CD68+	Stromal TAMs, CD68+	CD47
0	missing CD68 positive cells	missing CD68 positive cells	negative result (H-score = 0)
1	low concentration with a mean number of CD68 positive cells < 13	low concentration with a mean number of CD68 positive cells < 25	low score (H-score ≤ 74)
2	high concentration with a mean number of IT CD68 positive cells ≥ 13	high concentration, with a mean number of ST CD68 positive cells ≥ 25	high score (H-score > 74)

## Data Availability

The authors declare that all related data are available from the corresponding author upon reasonable request.

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
