# Peer review of "Integrated Analysis of Phagocytic and Immunomodulatory Markers in Cervical Cancer Reveals Constellations of Potential Prognostic Relevance"

_ijms, 2024, doi:10.3390/ijms25169117_

Round 1

Reviewer 1 Report

Comments and Suggestions for Authors

The manuscript by Yordanov et al., entitled “Integrated Analysis of Phagocytic and Immunomodulatory  Markers in Cervical Cancer Reveals Constellations of Potential Prognostic Relevance “ investigated that  the expression of CD68 and CD47 in patients  with different histological variant of CC at different stages and to find a relationship  between their expression and International Federation of Gynaecology and Obstet[1]rics stage, T and N status and surveillance. The authors find  that the level of CD68 infiltration is the histological variant of the tumor and the  highest level of  CD68  observed in SCC, while the high levels of CD47 expression were noted to be  more frequently in pT1B  than pT2A and pT2B. Infiltration by large numbers of CD68-IT cells was much more common among patients with high expression of CD47 on tumor cells. A high level of infiltration by CD68-ST cells was associated with a worse prognosis, and a high level of infiltration by CD68-ST cells with a lower risk of death from cancer. The manuscript is wel written, and the results are well presented , however, the a representative immune histochemistry  analysis  of different of both CD68 and CD47 in tumor stages will improve the quality of the study and make the study more interesting for the reader of the journal.

Comments on the Quality of English Language

Minor editing of English language required.

Author Response

Dear Reviewer,

We are deeply grateful for your comprehensive review. Thank you for your insightful and constructive review of our paper.

We incorporated the recommended changes. All incorporated changes are highlighted by using the Track and Changes in Word.

The manuscript by Yordanov et al., entitled “Integrated Analysis of Phagocytic and Immunomodulatory  Markers in Cervical Cancer Reveals Constellations of Potential Prognostic Relevance “ investigated that  the expression of CD68 and CD47 in patients  with different histological variant of CC at different stages and to find a relationship  between their expression and International Federation of Gynaecology and Obstet[1]rics stage, T and N status and surveillance. The authors find  that the level of CD68 infiltration is the histological variant of the tumor and the  highest level of  CD68  observed in SCC, while the high levels of CD47 expression were noted to be  more frequently in pT1B  than pT2A and pT2B. Infiltration by large numbers of CD68-IT cells was much more common among patients with high expression of CD47 on tumor cells. A high level of infiltration by CD68-ST cells was associated with a worse prognosis, and a high level of infiltration by CD68-ST cells with a lower risk of death from cancer. The manuscript is wel written, and the results are well presented , however, the a representative immune histochemistry  analysis  of different of both CD68 and CD47 in tumor stages will improve the quality of the study and make the study more interesting for the reader of the journal.

Author’s Reply: We agree with the reviewer. We changed table 3 and included the information about CD68 and CD47 expression in tumor stages. We incorporated the following test in the result section:

There is a significant difference in CD47 expression by FIGO stage (Fisher’s test p=0.0012) (Table 3).

And added the following test in the discussion section

This observation is also confirmed by the fact that CD47 expression is higher in FIGO stage I than stage II and III (p=0.0012).

We also added one more figure to fig 1 - 1C and change the whole fig 1.

Reviewer 2 Report

Comments and Suggestions for Authors

This study found that the only clinicopathological feature implicating the level of CD68 infiltration was the histological variant of the tumor and only for CD68-IT high levels were only observed in SCC. High levels of CD47 expression were seen more frequently in pT1B than pT2A and pT2B. Infiltration by large numbers of CD68-IT cells was much more common among patients with high expression of CD47 on tumor cells. A high level of infiltration by CD68-ST cells was associated with a worse prognosis, and a high level of infiltration by CD68-ST cells with a lower risk of death from cancer. It is recommended that Int. J. Mol. Sci. accept this article after the author revises and supplements the relevant data.

1. There are many research articles on the relationship between cervical cancer and CD68 and CD47. It is suggested that the author further sort out and compare in the discussion section to highlight the innovative points of this article.

2. The corresponding pathological sections should be provided for the immunohistochemistry part.

Comments on the Quality of English Language

Minor editing of English language required.

Round 2

Reviewer 2 Report

Comments and Suggestions for Authors

The requested revisions have been made by the authors, so that the article can be accepted.